# Exploring pooled analysis of pretested items to monitor the performance of medical students exposed to different curriculum designs

Pedro Tadao Hamamoto Filho[1]*, Pedro Luiz Toledo de Arruda Lourenção[2☯], Joélcio Francisco Abbade[3☯], Dario Cecílio-Fernandes[4☯], Jacqueline Teixeira Caramori[5‡], Angélica Maria Bicudo[6‡]

1 Department of Neurology, Psychology and Psychiatry, Botucatu Medical School, UNESP–Universidade Estadual Paulista, Botucatu, São Paulo, Brazil, 2 Department of Surgery and Orthopedics, Botucatu Medical School, UNESP–Universidade Estadual Paulista, Botucatu, São Paulo, Brazil, 3 Department of Gynecology and Obstetrics, Botucatu Medical School, UNESP–Universidade Estadual Paulista, Botucatu, São Paulo, Brazil, 4 Department of Medical Psychology and Psychiatry, School of Medical Sciences, UNICAMP–Universidade Estadual de Campinas, Campinas, São Paulo, Brazil, 5 Department of Internal Medicine, Botucatu Medical School, UNESP–Universidade Estadual Paulista, Botucatu, São Paulo, Brazil, 6 Department of Pediatrics, School of Medical Sciences, UNICAMP–Universidade Estadual de Campinas, Campinas, São Paulo, Brazil

☯ These authors contributed equally to this work.
‡ These authors also contributed equally to this work.
* pedro.hamamoto@unesp.br

## Abstract

Several methods have been proposed for analyzing differences between test scores, such as using mean scores, cumulative deviation, and mixed-effect models. Here, we explore the pooled analysis of retested Progress Test items to monitor the performance of first-year medical students who were exposed to a new curriculum design. This was a cross-sectional study of students in their first year of a medical program who participated in the annual inter-institutional Progress Tests from 2013 to 2019. We analyzed the performance of first-year students in the 2019 test and compared it with that of first-year students taking the test from 2013 to 2018 and encountering the same items. For each item, we calculated odds ratios with 95% confidence intervals; we also performed meta-analyses with fixed effects for each content area in the pooled analysis and presented the odds ratio (OR) with a 95% confidence interval (CI). In all, we used 63 items, which were divided into basic sciences, internal medicine, pediatrics, surgery, obstetrics and gynecology, and public health. Significant differences were found between groups in basic sciences (OR = 1.172 [CI95% 1.005 CI 1.366], p = 0.043) and public health (OR = 1.54 [CI95% CI 1.25–1.897], p < 0.001), which may reflect the characteristics of the new curriculum. Thus, pooled analysis of pretested items may provide indicators of different performance. This method may complement analysis of score differences on benchmark assessments.

**Data Availability Statement:** All relevant data are within the manuscript and its Supporting Information files.

**Funding:** PTHF and AMB have received an award from the National Board of Medical Examiners (PA, PA, USA). GRANT_NUMBER: Proposal LAG5-2020. https://contributions.nbme.org/about/latin-america-grants The funders had no role in study design, data collection and analysis, decision to publish, or preparation of the manuscript.

**Competing interests:** The authors have declared that no competing interests exist.

## Introduction

Over the last 30 years, several medical schools have implemented new undergraduate educational programs, which focus on early contact with patients, the inclusion of humanities sciences, and community-based approaches [1, 2]. Moreover, recognizing the value of problematization and multidisciplinary instruction, teaching methods have been reappraised [3, 4]. When a curriculum is changed, students, faculties, and curriculum managers need meaningful ways to ascertain that the new curriculum has improved upon the previous one [5].

Ultimately, improvements in students' subsequent professional performance and patient outcomes could serve as the best evidence for a curriculum's effectiveness. However, this information would not be easy to obtain due to the difficulties to establish a direct linkage between curriculum design, education quality, and health indicators [6, 7]. It is more feasible to measure students' knowledge, not only to gauge student performance and but also to identify gaps and strengths in a new curriculum [8–11].

In this sense, curriculum-based measurements (CBM) are helpful to assess students' progression and the effectiveness of the curriculum design [12]. CBM can be aided by benchmark assessments, which are periodic assessments of students' progress towards achieving their learning objectives. Benchmark assessment provides timely information, allowing adaptation of educational strategies for effective learning [13, 14], either at individual, school, and regional levels [15].

For cross-institutional comparison of student achievement, the Progress Test has been shown to be a possible tool, in medical education, as benchmark assessment if based on longitudinal data [16]. In curriculum comparisons, the Progress Test has been used in two ways: common exams given to different cohorts [11] or at different schools [17]; or different exams given to different cohorts (in this case, equations are necessary to avoid bias and scale scores) [18]. Several methods have been proposed for analyzing differences between scores, such as using mean scores [17], cumulative deviation [19], and mixed-effect models [20]. However, since different progress tests may have different levels of difficulty, mean scores may not allow for reliable comparisons between scores on different exams because exams may have different levels of difficulty. Cumulative deviation, meanwhile, requires a longitudinal appraisal of exams' standard deviation, making it difficult to estimate conclusions based on single-point tests. Moreover, when comparing different cohorts, these methods rely on different items, and this approach may threaten validity and reliable comparison. Finally, these methods only allow for comparisons at the group level.

Using a method that allows comparisons between the same items may overcome these challenges, and a method that allows for comparing individual items may provide richer information on specific knowledge gaps. Here, we explore a statistical method which can be applied to assess the effectiveness of curricular change.

## Methods

### Study setting and participants

This analysis used data from the Interinstitutional Progress Test, in which the students of Botucatu Medical School, Universidade Estadual Paulista (BMS-UNESP), participate since 2005 [21]. This study was approved by the local institutional review board. Written consent from the students was not necessary because this study dealt with an anonymized database with aggregated information. This cross-sectional study was conducted at BMS-UNESP, in Botucatu, São Paulo State, Brazil. We included students in the first year of the medical program at BMS-UNESP who had participated in the Interinstitutional Progress Test from 2013 to 2019.

In Brazil, an undergraduate medical course takes 6 years [22]. Like the majority of the Brazilian schools, the original medical program curriculum at BMS-UNESP was divided into three cycles: basic sciences (1st and 2nd years), clinical sciences (3rd and 4th years), and the clerkship (5th and 6th years). Subject-based teaching was used for the basic sciences, which were organized into traditional subjects (e.g., anatomy, biochemistry, physiology, microbiology, and immunology). In 2019, the new curriculum was implemented, consisting of two cycles: the pre-clinical cycle (1st to 3rd years) and the clerkship (4th to 6th years). In the new curriculum, basic sciences were taught using a systems-based approach (e.g., traditional subjects of cellular biology, biochemistry, hematology were integrated into a course on the "cell," while neuroanatomy, physiology, embryology, and neurology were integrated into a course on the "nervous system"). Moreover, social sciences-related disciplines (e.g., epidemiology, sociology, public health), which were previously taught independently, were organized along a structured axis for the humanities, including interdisciplinary and community-based approaches.

In this study, we analyzed first-year medical students' performance on the 2019 annual IPT and compared it with that of first-year students taking the exam from 2013 to 2018 and encountering the same items.

## Progress Test

Progress Test is a longitudinal assessment that measures students' knowledge on subsequent yet different tests. Through first to last year of medical training, all students answer the same test and receive feedback on their performance. The Progress Test is based on a blueprint with questions requiring both a lower and a higher level of cognitive processing, covering the content that every just-graduated student should have [23, 24].

As is common in Brazilian schools, the Progress Test is given once a year for formative purposes: each student takes the test each year throughout the undergraduate course and the students' performance does not affect student advancement decisions. BMS is one of the main public schools in São Paulo state, and it formed a consortium with other medical schools to prepare and administer the state's annual interinstitutional progress test.

The consortium of schools develops the annual IPT using only new items; these conform to a fixed blueprint and cover six content areas: basic sciences, internal medicine, surgery, pediatrics, obstetrics and gynecology, and public health (20 items per area for a total of 120 items). Items are multiple choice questions with four options and a single correct answer. Preferably, the items are clinical vignette-based aiming for applied knowledge rather than knowledge recall [25]. In 2019, four existing consortia in São Paulo state developed the exam by selecting the best pre-tested items (tested between 2013 and 2018 and preferably, between 2016 and 2018, with good discrimination indices) conforming to the commonly used blueprint. The use of pre-tested items in 2019 allowed us to compare the performance of the different groups of first-year students (the first-year students in 2019 versus the first-year students from 2013 to 2018) on the items.

Importantly, as the students' performance does not affect advancement decisions, the students do not study for the test, and item-sharing between the cohorts does not occur.

## Statistical analysis

Statistical analyses were performed using MedCalc for Windows, version 19.4 (MedCalc Software, Ostend, Belgium). We presented correct answers as item counts with percentages for each group. Furthermore, we calculated odds ratios (ORs) with a 95% confidence interval (CI) for each item, and we performed meta-analyses with fixed effects for each content area in the pooled analysis (presenting them as OR with a 95% CI). The statistical significance was set at

an alpha of 0.05, and $I^2$ statistics were used to assess the heterogeneity among the items' results ($I^2$ values of 25%, 50%, and 75% are interpreted as representing small, moderate, and high levels of heterogeneity) [26].

## Results

Of the 120 items on the full exam, 63 were from our consortium and were, therefore, eligible for comparison. These items were divided into the following categories: 17 from basic sciences; 11 from surgery; 9 from internal medicine, obstetrics and gynecology, and public health; and 8 from pediatrics. Regarding the years when items were previously used, 20, 16, and 21 items were from 2016, 2017, and 2018, respectively. Six other items were from 2013 and 2014 (two in 2013 and four in 2014), all from the basic sciences. Table 1 summarizes these data as well as the number of students who took the exam each year.

Among the six content areas, significant differences were found for basic sciences and public health. In the other four content areas (internal medicine, pediatrics, surgery, and obstetrics and gynecology), there were no differences in performance between the 2019 students and their counterparts in earlier years (Fig 1).

In the basic sciences, the pooled analysis showed that the 2019 students had superior performance (OR = 1.172 [CI95% 1.005 CI 1.366], p = 0.043). Among the 17 items, the 2019 students' performance was statistically different on four items: superior on three items and inferior on one.

In public health, the 2019 students' performance was also superior (OR = 1.54 [CI95%: CI 1.25–1.897], p < 0.001). The difference was weighted for superior performance on 3/9 items. On one item (relating to epidemiology), the OR reached 7.00 (CI95%: CI 3.62–13.55). When this item appeared on the earlier test, 36.67% of the students answered correctly; in 2019, this percentage increased to 80.21%.

In internal medicine, there were differences on two items: one result favored the new curriculum and the other, the former curriculum. However, the pooled analysis showed no statistically significant difference: OR = 0.93 (CI95% CI 0.735–1.176, p = 0.544).

In pediatrics, no item showed a significant difference, nor did the pooled analysis: OR = 1.128 (CI95% CI 0.886–1.435, p = 0.329).

Among the 11 items from surgery, the students exposed to the new curriculum performed better on one item and worse on three. The pooled analysis showed no difference: OR = 1.015 (CI95% CI 0.725–1.421, p = 0.765).

Finally, in obstetrics and gynecology, the 2019 students' performance was better on two items and worse on one item, with the pooled analysis failing to show any significant difference: OR = 1.164 (CI95% CI 0.944–1.436, p = 0.154).

**Table 1. Distribution of the number of students and number of items previously tested.**

|  | 2013 | 2014 | 2016 | 2017 | 2018 | Total |
|---|---|---|---|---|---|---|
| Number of students | 88 | 96 | 90 | 95 | 90 |  |
| Number of items | 2 | 4 | 20 | 16 | 21 | 63 |
| Basic sciences | 2 | 4 | 4 | 4 | 3 | 17 |
| Internal medicine | 0 | 0 | 4 | 1 | 4 | 9 |
| Pediatrics | 0 | 0 | 1 | 2 | 5 | 8 |
| Surgery | 0 | 0 | 2 | 5 | 4 | 11 |
| Obstetrics & gynecology | 0 | 0 | 3 | 3 | 3 | 9 |
| Public health | 0 | 0 | 6 | 1 | 2 | 9 |

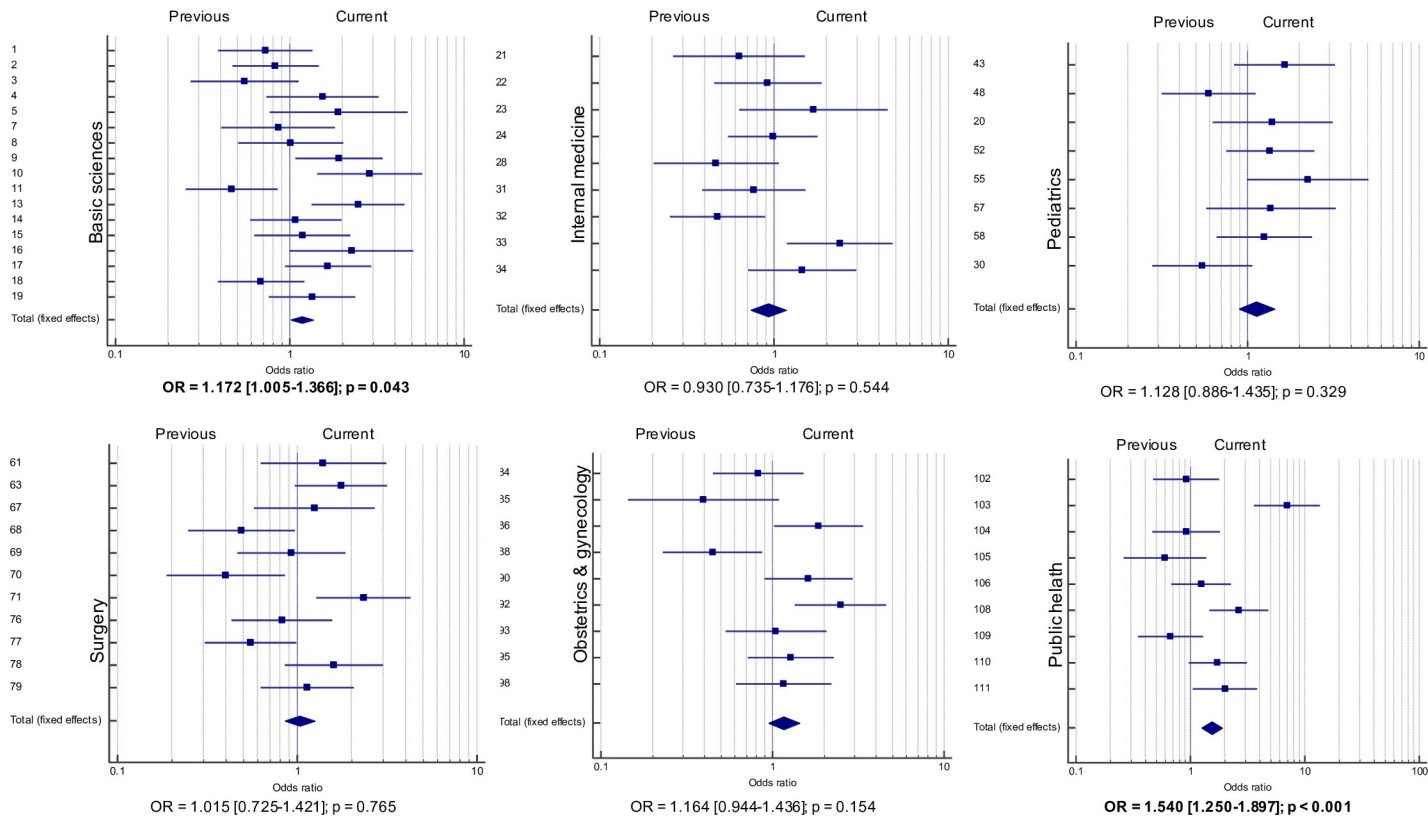

**Fig 1. Forest plots of the pooled analysis according to the exam's different content areas.** The vertical axis represents the item number on the 2019 exam. Each point represents the OR with its respective CI (horizontal bars). When points appear on the left of the vertical line identified as "1", it indicates that students exposed to the old curriculum performed better, whereas points on the right indicate better performance for students with the new curriculum. Significant differences were found in basic sciences and public health.

The heterogeneity analysis of the items showed high percentages of variation across the items, with $I^2$ values ranging from 44.47% (Pediatrics) to 80.26% (Public Health). Table 2 shows the $I^2$ values for each content area.

## Discussion

Pooled analysis is commonly used in meta-analyses and systematic reviews of clinical trials to summarize the scientific evidence provided by single studies [27]. In medical education, similar approaches have been adopted to estimate the effects of specific education interventions that are linked by a common objective [28]. However, we did not find any evidence of previous pooled analyses based on items from benchmark assessments. In this exploratory study, we

**Table 2. $I^2$ statistics for heterogeneity evaluation in the six content areas of the exam.**

| Content Area | $I^2$ value (%) | CI 95% | p value |
|---|---|---|---|
| Basic sciences | 61.79 | 35.33–77.42 | 0.0004 |
| Internal medicine | 55.90 | 7.00–79.09 | 0.0202 |
| Pediatrics | 47.44 | 0.00–76.63 | 0.0647 |
| Surgery | 64.92 | 33.25–81.56 | 0.0015 |
| Obstetrics & gynecology | 65.73 | 30.36–83.14 | 0.0029 |
| Public health | 80.26 | 63.31–89.38 | < 0.0001 |

delineated how this approach can be used to compare student performance and can serve as an alternative to single-point comparisons, though it cannot reduce the limitation of the low reliability of single measures.

Our results suggest that students exposed to the new curriculum performed better than their old-curriculum counterparts in basic sciences and public health. Conversely, no significant differences were observed in the applied clinical sciences. These results should be interpreted with caution, as the majority of content areas showed moderate to high heterogeneity. This means that a large proportion of the variation in the observed estimates was due to heterogeneity across the items in the analysis [29, 30], which may be related to the small sample size in the present case.

In cases of high heterogeneity, the qualitative appraisal of results is important to understand the estimated effect. In this regard, the philosophy of the new curriculum (i.e., more integration between the basic sciences and social sciences applicable to medicine) may support the performance of students in basic sciences and public health. Notably, students exposed to the new curriculum integrating basic and clinical sciences may be better prepared for the interinstitutional Progress Test [31], which uses high taxonomy vignette-based items [25]. In addition, experiences in community settings may also contribute to early medical education [32]. Together, these curriculum designs may decisively contribute to better educational outcomes [33].

Accordingly, no performance differences were observed in items related to applied clinical sciences (internal medicine, pediatrics, surgery, obstetrics, and gynecology). This is probably because in both the old and new curricula, students had little exposure to these areas (particularly to the diagnosis and treatment of diseases). Therefore, it is understandable that we detected a greater impact on student performance in basic sciences and public health, as these were content areas that were more substantially changed in the new curriculum.

Our study has some limitations that should be mentioned. First, the comparison group is not uniform. The "control" group comprised different cohorts of students, and specific characteristics of student groups may have introduced noise into the results. However, other than the curriculum change, no other institutional changes can explain the detected differences. Second, our sample has half the items of a full exam (63/120 items), and it is known that when using fewer items, reliability is not guaranteed [34]. Third, the Progress Test itself is too brief and covers a broad range of content areas. Thus, obtaining accurate indices of performance in individual content sub-areas is difficult, as the testing of these areas is based only on a few items [35]. Finally, we did not use other tools for the curriculum comparison and, therefore, the superiority of the new one is not unequivocal.

However, as stated previously, this is an exploratory study showing the possible use of pooled analysis to compare performance. Our study provides a blueprint for how other investigations might use a similar approach to evaluate programmatic changes in educational settings. This method may be especially useful for: 1) detecting significant differences on tests that employ repeated items; 2) comparing performance at different institutions that use a same test; 3) reporting the performance of students on benchmark assessments, beyond the Progress Test. It is not expected, however, that this method will completely replace other tools. Rather, it can complement the full set of possibilities.

Moreover, the comparison of each item may be important to detect knowledge gaps among the students, even when the pooled analysis shows no difference. Further studies may address this point and set comparisons between different statistical procedures.

In conclusion, pooled analysis of pretested items can be a statistical method to assess the effectiveness of curricular changes.

## Supporting information

**S1 Data. Correct answers with percentages for each item and group, and odds ratio calculation.**
(XLSX)

## Author Contributions

**Conceptualization:** Pedro Tadao Hamamoto Filho.

**Data curation:** Pedro Tadao Hamamoto Filho.

**Formal analysis:** Pedro Tadao Hamamoto Filho, Pedro Luiz Toledo de Arruda Lourenção, Joélcio Francisco Abbade.

**Funding acquisition:** Pedro Tadao Hamamoto Filho, Angélica Maria Bicudo.

**Investigation:** Pedro Tadao Hamamoto Filho, Joélcio Francisco Abbade.

**Methodology:** Pedro Tadao Hamamoto Filho, Pedro Luiz Toledo de Arruda Lourenção, Joélcio Francisco Abbade.

**Project administration:** Pedro Tadao Hamamoto Filho.

**Supervision:** Jacqueline Teixeira Caramori, Angélica Maria Bicudo.

**Validation:** Dario Cecílio-Fernandes.

**Writing – original draft:** Pedro Tadao Hamamoto Filho.

**Writing – review & editing:** Pedro Luiz Toledo de Arruda Lourenção, Joélcio Francisco Abbade, Dario Cecílio-Fernandes, Jacqueline Teixeira Caramori, Angélica Maria Bicudo.

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
