## [Decision Letter · Decision Letter 0]

16 Jun 2021

PONE-D-20-40753

Using pooled analysis of pretested items from the Progress Test to monitor performance of first-year medical students exposed to different curriculum designs: an exploratory study

PLOS ONE

Dear Dr. Hamamoto Filho,

Thank you for submitting your manuscript to PLOS ONE. After careful consideration, we feel that it has merit but does not fully meet PLOS ONE’s publication criteria as it currently stands. Therefore, we invite you to submit a revised version of the manuscript that addresses the points raised during the review process.

I highly recommend you address the following. 

1. Clarify the use of the assessment as a benchmark assessment rather than a progress test and the overall goal of the study. 2. Clarify more specifically the assessment question objectives and how these relate to the overall heterogeneity. 3. Describe whether any other measures were used to compare the curriculum. As was indicated in the discussion, a single measure can yield low reliability.

We look forward to receiving your revised manuscript.

Kind regards,

Amy Prunuske

Academic Editor

PLOS ONE

Additional Editor Comments (if provided):

Thank you for your submissions. The reviewers valued your work and have provided a number of suggestions to improve the manuscript.

I highly recommend you clarify the following.

1. Clarify the use of the assessment as a benchmark assessment rather than a progress test and the overall goal of the study.

2. Please clarify more specifically the assessment question objectives and how these relate to the overall heterogeneity.

3. Describe whether any other measures were used to compare the curriculum. As was indicated in the discussion, a single measure can yield low reliability.

Journal Requirements:

Reviewers' comments:

Reviewer's Responses to Questions

**Comments to the Author**

1. Is the manuscript technically sound, and do the data support the conclusions?

Reviewer #1: Partly

Reviewer #2: Yes

Reviewer #3: Yes

2. Has the statistical analysis been performed appropriately and rigorously? 

Reviewer #1: I Don't Know

Reviewer #2: Yes

Reviewer #3: Yes

3. Have the authors made all data underlying the findings in their manuscript fully available?

Reviewer #1: Yes

Reviewer #2: Yes

Reviewer #3: Yes

4. Is the manuscript presented in an intelligible fashion and written in standard English?

Reviewer #1: Yes

Reviewer #2: Yes

Reviewer #3: Yes

5. Review Comments to the Author

Reviewer #1: This paper can make a worthwhile contribution to medical education literature but in it’s current form, it has several flaws:

1. My understanding of progress testing is that it involves repeated testing of the same cohort. This study seems to report on testing of subsequent year 1 cohorts over several years and therefore better fits the definition of benchmarking assessment rather than progress testing per se.

2. The introduction largely focuses on progress testing as an assessment methodology but the discussion and the value of the study seem to be more about the use of pooled analysis for comparison of performance between cohorts and curricula. I wonder if in fact, the focus of this paper needs to shift away from progress testing and towards the statistical methods which can be applied to assess the effectiveness of curricula change.

3. The re-use of questions in subsequent years can be problematic in that students recall and share questions and they become available for subsequent cohorts. Analysis of the performance of repeated items across years should be conducted to identify whether this has occurred in this case. Further, the authors could provide more information about the context of the different medical schools to offer readers a clearer impression of how sharing may or may not occur between them.

4. Data was collected using the interinstitutional progress test from 2005 – why is only data from 2013 used here?

5. Apart from performance in the assessment, did the cohorts differ in other ways? Are demographic data available to describe the cohorts?

Reviewer #2: This manuscript provides a sound example of how pooled analyses of a standardized examination, the Progress Test, can be used to retrospectively evaluate impact of curricular change in medical education. While the findings are limited due to the smaller sample size and high degree of heterogeneity, the authors have done an excellent job of demonstrating how this approach was used, interpreted, and the strengths / limitations of it compared to a single-point comparison analysis. Furthermore, the use of a smaller sample size makes this work relatable to many and provides a template for how others might proceed under similar constraints. Though the work itself isn't ground-breaking, it does provide a very accessible (well written and clearly described) blueprint for how other investigators might use a similar approach to evaluate programmatic change at their own institutions. For this reason, this manuscript offers substantial practical application value to its readers.

Reviewer #3: SUMMARY OF THE RESEARCH

In this manuscript, the authors describe a method of pooled item analysis to evaluate learning in several disciplines between two curricula. Overall the writing was clear, making it easy to follow the results and conclusions of the study. The “old” curriculum consisted of three phases (Basic Science M1-M2, Clinical Science M3-M4, Clerkship M5-M6) where the basic sciences were presented in subject-based courses. The “new” curriculum is divided into two phases (Preclinical M1-M3, Clerkship M4-M6) where the basic sciences are taught within systems-based units. The authors used a subset of 63 questions from the previously published Progress Test. All questions were used on the Progress Test at some point between 2013-2018, and student performance on those questions at those times was used to assess learning in the “old” curriculum. Students in 2019 who had begun the “new” curriculum answered the same 63 questions and their performance was compared to that in the “old” curriculum. The authors compared performance across curricula for each question, but also used a pooled analysis to evaluate differences in students understanding within a content area (ex: basic sciences, public health, surgery). Results showed improved performance on basic sciences and pubic health questions with the new curriculum compared to the old curriculum. I2 analysis showed a high degree of heterogeneity. Study was done well given the data available, but would have liked more description of specific changes to the curriculum that led to improvements, though that was beyond the scope of the study. Overall, it seems that pooled analysis of items proved a useful analytical tool to compare student performance across curricula.

Recommendation: Accept with minor revisions.

DISCUSSION OF SPECIFIC AREAS FOR IMPROVEMENT

Introduction

1. Need reference for content in lines 48-50.

2. Lines 55-57 – please clarify what is meant by “easy to obtain”. It seems the authors mean it is not time efficient to evaluate long-term metrics like clinical performance and assessments of new curricula need to be done in real time to evaluate current learning.

3. Lines 60-65 – please describe the types of questions posed in the Progress Test – are they clinical vignettes? Are they assessing low-level knowledge?

4. Line 61 - what is meant by “repeated measures”? Do students take the test each year of their training)?

Materials & Methods

1. Please describe the typical path of medical training in Brazil. Do all students spend 6 years in medical school? Is the format of your new curriculum the norm there?

2. Study setting and participants

a. Describe any exclusion criteria.

b. Describe changes to the content of the curriculum. The authors mention the social sciences disciplines were taught differently, but does this equate to inclusion of more content or just redistribution of content?

c. Why were first year students used for this study when the first phase lasts at least two years for both curricula?

d. Was there a difference in when certain content was covered in the old and new curriculum relative to when students took this test?

3. Progress Test

a. Line 110 – clarify if this means each student takes the test each year throughout their training

b. The reader needs more context here, please describe types of questions included in this test

4. Line 131 – describe interpretation of I2 analysis

Results

1. Why were there no questions selected from 2015?

2. It seems that including questions from so many different years increases the noise of your sample, why not select questions from a smaller subset of years?

3. Have the authors evaluated student performance on these questions in the old vs. new curriculum when students were M2s?

4. Line 151-153 – this needs clarification. When points appear to the right or left of what? 1?

5. Line 162-164 – this is interesting, what was the topic and what differed in the new curriculum that improved understanding of this topic so dramatically?

6. Line 175 – place the p value inside the parenthesis to match formatting of the other paragraphs

7. Issue with formatting for lines 181-186 (seems related to the editorial software)

Discussion

1. Line 198-200 – is this what would be expected given changes to the curriculum?

2. Line 220-222 – it is concerning that the “control” group consists of multiple cohorts while the “new” group is all within a single cohort. It seems this would introduce a decent amount of noise into the data.

3. It is unclear if the goal of the study is to demonstrate utility of pooled analysis as a method or demonstrate that their new curriculum improved student understanding of basic sciences and public health.

ADDITIONAL POINTS

The comparison of student performance in the old vs. new curriculum is valuable for quality assurance at their institution. It is unclear how novel a finding this is. One would hope that after a major curriculum redesign, improved learning could be shown, but it is unclear where to attribute those improvements. What did they do to improve learning in the basic sciences and public health? Do they have recommendations for other educators? It seems the major conclusion the authors are trying to show is that pooled analysis of items is a useful method. However, the data seems very noisy and difficult to draw conclusions from.

6. PLOS authors have the option to publish the peer review history of their article (what does this mean?). If published, this will include your full peer review and any attached files.

Reviewer #1: No

Reviewer #2: No

Reviewer #3: No

---

## [Author Response · Author response to Decision Letter 0]

28 Jul 2021

Dear Editor,

 We thank you and the reviewers for the thoughtful comments on our manuscript. We are sure that they helped us improving its quality and we hope that we have successfully addressed all the concerns and suggestions.

 Bellow, we answer each of the comments.

Editor:

I highly recommend you address the following. 

1. Clarify the use of the assessment as a benchmark assessment rather than a progress test and the overall goal of the study.

 We put emphasis on the possible use of the proposed method for benchmark assessment. Therefore, the goal of the study was the presentation of the method and the curriculum comparison was used to exemplify the pooled analysis potential.

2. Clarify more specifically the assessment question objectives and how these relate to the overall heterogeneity.

We provided more information about the Progress Test, its design and characteristics, supporting more information about the items. We believe that the heterogeneity may be related to the sample size – this should be tested in next studies, since here, our attempt is to present and explore the use of this measurement.

3. Describe whether any other measures were used to compare the curriculum. As was indicated in the discussion, a single measure can yield low reliability.

No other measures were used to compare the curriculums. Since the reviewers asked us to shift the focus of the paper to the proposed method, we believe that more details on curriculum comparisons are not needed in this paper. But we strongly agree that for a better comparison, other tools would be necessary. We stated this point in the Limitations section.

Reviewer #1:

This paper can make a worthwhile contribution to medical education literature but in it’s current form, it has several flaws:

We thank the Reviewer for the good observations made. We believe that our study may be of help to medical education and we hope that the flaws were now overcome.

1. My understanding of progress testing is that it involves repeated testing of the same cohort. This study seems to report on testing of subsequent year 1 cohorts over several years and therefore better fits the definition of benchmarking assessment rather than progress testing per se.

We agree with the Reviewer. The comparison of repeated items resembles benchmark assessment. We made changes throughout the manuscript. However, each of the year 1 students were administered different tests (Progress Tests). The 2019 year 1 students were administered a test whose items were used in previous tests. Besides, since the items used came from the Progress Test, it was important to keep some descriptions about it. 

2. The introduction largely focuses on progress testing as an assessment methodology but the discussion and the value of the study seem to be more about the use of pooled analysis for comparison of performance between cohorts and curricula. I wonder if in fact, the focus of this paper needs to shift away from progress testing and towards the statistical methods which can be applied to assess the effectiveness of curricula change.

We made changes in the Title, Introduction and Discussion to emphasize the statistical method that is proposed. Accordingly, information and references related to the Progress Test were suppressed.

3. The re-use of questions in subsequent years can be problematic in that students recall and share questions and they become available for subsequent cohorts. Analysis of the performance of repeated items across years should be conducted to identify whether this has occurred in this case. Further, the authors could provide more information about the context of the different medical schools to offer readers a clearer impression of how sharing may or may not occur between them.

 The problem of recall is not a point in this study because the intervention group (new curriculum) has never before undergone the test. Sharing questions could be a problem but, actually, as we observe for more than 15 years, this does not occur because students do not have to study for the test (and really do not study). The test book is given to the students after the its administration and even senior students exposed eventual repeated questions cannot recall these questions. We stated this point in the text.

4. Data was collected using the interinstitutional progress test from 2005 – why is only data from 2013 used here?

Actually, 2005 is the year when the school started administering the Progress Test. For the elaboration of the 2019 test, there were included only items from 2013 to 2018 (preferably 2016 to 2018). This was a decision of the Interinstitutional Progress Test commission.

5. Apart from performance in the assessment, did the cohorts differ in other ways? Are demographic data available to describe the cohorts?

 Since we dealt with aggregate information of the tests, we do not have demographic information. However, there was no particular change in 2019 admission policies of the school that could lead to differences between the cohorts.

Reviewer #2:

This manuscript provides a sound example of how pooled analyses of a standardized examination, the Progress Test, can be used to retrospectively evaluate impact of curricular change in medical education. While the findings are limited due to the smaller sample size and high degree of heterogeneity, the authors have done an excellent job of demonstrating how this approach was used, interpreted, and the strengths / limitations of it compared to a single-point comparison analysis. Furthermore, the use of a smaller sample size makes this work relatable to many and provides a template for how others might proceed under similar constraints. Though the work itself isn't ground-breaking, it does provide a very accessible (well written and clearly described) blueprint for how other investigators might use a similar approach to evaluate programmatic change at their own institutions. For this reason, this manuscript offers substantial practical application value to its readers.

We thank the Reviewer for appreciating our study. Indeed, we are quite aware of the limitations that it has, including the small sample size. However, we do believe that the proposed approach may have a broad audience and use in other institutions.

Reviewer #3: 

SUMMARY OF THE RESEARCH

In this manuscript, the authors describe a method of pooled item analysis to evaluate learning in several disciplines between two curricula. Overall the writing was clear, making it easy to follow the results and conclusions of the study. The “old” curriculum consisted of three phases (Basic Science M1-M2, Clinical Science M3-M4, Clerkship M5-M6) where the basic sciences were presented in subject-based courses. The “new” curriculum is divided into two phases (Preclinical M1-M3, Clerkship M4-M6) where the basic sciences are taught within systems-based units. The authors used a subset of 63 questions from the previously published Progress Test. All questions were used on the Progress Test at some point between 2013-2018, and student performance on those questions at those times was used to assess learning in the “old” curriculum. Students in 2019 who had begun the “new” curriculum answered the same 63 questions and their performance was compared to that in the “old” curriculum. The authors compared performance across curricula for each question, but also used a pooled analysis to evaluate differences in students understanding within a content area (ex: basic sciences, public health, surgery). Results showed improved performance on basic sciences and public health questions with the new curriculum compared to the old curriculum. I2 analysis showed a high degree of heterogeneity. Study was done well given the data available, but would have liked more description of specific changes to the curriculum that led to improvements, though that was beyond the scope of the study. Overall, it seems that pooled analysis of items proved a useful analytical tool to compare student performance across curricula.

Recommendation: Accept with minor revisions.

We thank the reviewer for the meticulous observations made on our manuscript. Certainly, the suggestions improved the quality of the report. We hope that we have successfully addressed all the raised questions.

DISCUSSION OF SPECIFIC AREAS FOR IMPROVEMENT

Introduction

1. Need reference for content in lines 48-50.

References were added.

2. Lines 55-57 – please clarify what is meant by “easy to obtain”. It seems the authors mean it is not time efficient to evaluate long-term metrics like clinical performance and assessments of new curricula need to be done in real time to evaluate current learning.

We mean that it is difficulty to establish a direct linkage between curriculum design, education quality, and health indicators. This explanation was added to the text.

3. Lines 60-65 – please describe the types of questions posed in the Progress Test – are they clinical vignettes? Are they assessing low-level knowledge?

The majority of the questions are clinical vignette-based targeting to high taxonomic levels. This information was added in the Methods section because the Introduction was reformulated according to other Reviewer’s suggestions.

4. Line 61 - what is meant by “repeated measures”? Do students take the test each year of their training)?

Yes, Progress Test is a longitudinal assessment that measures students’ knowledge on subsequent yet different tests. Through first to last year of medical training, all students answer the same test. Again, this information was added in the Methods section due to rewriting of the Introduction.

Materials & Methods

1. Please describe the typical path of medical training in Brazil. Do all students spend 6 years in medical school? Is the format of your new curriculum the norm there?

In Brazil, the undergraduate medical course lasts 6 years. The 2-2-2 curriculum format (basic sciences, clinical sciences, and clerkship) is the most common design, either for traditional (lecture-based) or problem-based learning approaches. This new design (3-3) is innovative in the country. Brief information about it was provided.

2. Study setting and participants

a. Describe any exclusion criteria.

We excluded data from students that did not sit the test. However, this information is a mirror of the “inclusion criteria”, and, therefore, does not require detailing. Besides, since we dealt with aggregate data, no further “exclusion criteria” would be necessary.

b. Describe changes to the content of the curriculum. The authors mention the social sciences disciplines were taught differently, but does this equate to inclusion of more content or just redistribution of content?

We added the information that this change included interdisciplinary and community-based approaches. Since the Editor and other Reviewer suggested a clearer emphasis on the goal of the study (i.e., the pooled analysis), we believe that a very detailed description of the content changes is no longer needed.

c. Why were first year students used for this study when the first phase lasts at least two years for both curricula?

Because the test with repeated items was administered only once, in the same year in which the new curriculum was adopted, i.e., we did not have second-years students in the new curriculum so far. The repetition of items is uncommon because each year, new items are written.

d. Was there a difference in when certain content was covered in the old and new curriculum relative to when students took this test?

Again, since the Editor and other Reviewer suggested a clearer emphasis on the goal of the study (i.e., the pooled analysis), we believe that a very detailed description of the content changes is no longer needed.

3. Progress Test

a. Line 110 – clarify if this means each student takes the test each year throughout their training

 Yes, Progress Test is a longitudinal assessment on subsequent yet different tests. Through first to last year of medical training, all students answer the same test. In our school, the test is administered once a year. This information was added to the section.

b. The reader needs more context here, please describe types of questions included in this test

 We added more information regarding the Progress Test, as well as it’s questions.

c. Line 131 – describe interpretation of I2 analysis

 We added reference values for I2 analysis.

4. Results

a. Why were there no questions selected from 2015?

The committee of Progress Test opted not to use questions from 2015 because in this year, the test was a national exam (reference 21) with some differences from our pattern.

b. It seems that including questions from so many different years increases the noise of your sample, why not select questions from a smaller subset of years?

Yes, we are quite aware that so many years make the “control group” very noisy. If we use only a subset of years, we would have fewer items to compare. We agree that this is a limitation of our study and we stated it through the text. However, the purpose of study is to explore the new statistical approach.

c. Have the authors evaluated student performance on these questions in the old vs. new curriculum when students were M2s?

No. At the time that the test was administered, we did not have M2 students in the new curriculum (see 2c).

d. Line 151-153 – this needs clarification. When points appear to the right or left of what? 1?

Yes, left or right of the vertical axis of “1”. We clarified it in the figure legend.

e. Line 162-164 – this is interesting, what was the topic and what differed in the new curriculum that improved understanding of this topic so dramatically?

This item addressed an epidemiology content, that is better explored in the 1st year curriculum. We added the information in the text. However, without further details, because we have been asked by the Editor and other Reviewer to focus in the method of comparison, rather than in the curriculum change.

f. Line 175 – place the p value inside the parenthesis to match formatting of the other paragraphs

Thank you for the observation.

g. Issue with formatting for lines 181-186 (seems related to the editorial software)

Corrected.

5. Discussion

a. Line 198-200 – is this what would be expected given changes to the curriculum?

We did not have other indicators to have a well-defined expectation. This was the first measure comparing the curricula. However, surely, the academic community expects that the new curriculum functions better, but we believe that this discussion goes beyond the main purpose of this manuscript.

b. Line 220-222 – it is concerning that the “control” group consists of multiple cohorts while the “new” group is all within a single cohort. It seems this would introduce a decent amount of noise into the data.

We agree with the Reviewer and this was the reason why we recognized this limitation. However, as we shifted the focus of the manuscript on the pooled analysis.

c. It is unclear if the goal of the study is to demonstrate utility of pooled analysis as a method or demonstrate that their new curriculum improved student understanding of basic sciences and public health.

This point is in line with other comments. Therefore, we emphasized the utility of the pooled analysis. Thank you for the suggestion

6. ADDITIONAL POINTS

 The comparison of student performance in the old vs. new curriculum is valuable for quality assurance at their institution. It is unclear how novel a finding this is. One would hope that after a major curriculum redesign, improved learning could be shown, but it is unclear where to attribute those improvements. What did they do to improve learning in the basic sciences and public health? Do they have recommendations for other educators? It seems the major conclusion the authors are trying to show is that pooled analysis of items is a useful method. However, the data seems very noisy and difficult to draw conclusions from.

 Yes, it is very difficult to state that the new curriculum is directly related to the students’ better performance. We added some comments about it in the Discussion. And yes, the major conclusion is the utility of the pooled analysis. We made it clear throughout the text.

---

## [Decision Letter · Decision Letter 1]

31 Aug 2021

Exploring pooled analysis of pretested items to monitor the performance of medical students exposed to different curriculum designs

PONE-D-20-40753R1

Dear Dr. Hamamoto Filho,

We’re pleased to inform you that your manuscript has been judged scientifically suitable for publication and will be formally accepted for publication once it meets all outstanding technical requirements.

Kind regards,

Amy Prunuske

Academic Editor

PLOS ONE

Additional Editor Comments (optional):

Reviewers' comments:

Reviewer's Responses to Questions

**Comments to the Author**

1. If the authors have adequately addressed your comments raised in a previous round of review and you feel that this manuscript is now acceptable for publication, you may indicate that here to bypass the “Comments to the Author” section, enter your conflict of interest statement in the “Confidential to Editor” section, and submit your "Accept" recommendation.

Reviewer #4: (No Response)

2. Is the manuscript technically sound, and do the data support the conclusions?

Reviewer #4: Yes

3. Has the statistical analysis been performed appropriately and rigorously? 

Reviewer #4: Yes

4. Have the authors made all data underlying the findings in their manuscript fully available?

Reviewer #4: Yes

5. Is the manuscript presented in an intelligible fashion and written in standard English?

Reviewer #4: (No Response)

6. Review Comments to the Author

Reviewer #4: The authors present a pooled item analysis to evaluate learning in several disciplines between two curricula. Considering the available data - which are weak from the point of view of the demographic information of the respondents and in terms of sample size - the authors did a good job and the analysis provides useful information.

7. PLOS authors have the option to publish the peer review history of their article (what does this mean?). If published, this will include your full peer review and any attached files.

Reviewer #4: No

---

## [Editor Report · Acceptance letter]

2 Sep 2021

PONE-D-20-40753R1 

Exploring pooled analysis of pretested items to monitor the performance of medical students exposed to different curriculum designs 

Dear Dr. Hamamoto Filho:

I'm pleased to inform you that your manuscript has been deemed suitable for publication in PLOS ONE. Congratulations! Your manuscript is now with our production department. 

Kind regards, 

on behalf of

Dr. Amy Prunuske 

Academic Editor

PLOS ONE